# The Role of Gut Microbiome in Psoriasis: Oral Administration of *Staphylococcus aureus* and *Streptococcus danieliae* Exacerbates Skin Inflammation of Imiquimod-Induced Psoriasis-Like Dermatitis

**DOI:** 10.3390/ijms21093303

**Published:** 2020-05-07

**Authors:** Karin Okada, Yoshiaki Matsushima, Kento Mizutani, Keiichi Yamanaka

**Affiliations:** Department of Dermatology, Mie University Graduate School of Medicine, 2-174 Edobashi, Tsu, Mie 514-8507, Japan; okadakarin@clin.medic.mie-u.ac.jp (K.O.); matsushima-y@clin.medic.mie-u.ac.jp (Y.M.); k-mizutani@clin.medic.mie-u.ac.jp (K.M.)

**Keywords:** microbiome, psoriasis, inflammatory skin disease, *Staphylococcus aureus*, *Streptococcus danieliae*

## Abstract

Psoriasis is one of the common chronic inflammatory skin diseases in which inflammatory cytokines such as IL-17 and TNF-α play critical roles. Skin microbiome of psoriasis patients is reported to have elevated Staphylococcus and Streptococcus genus. There are controversial reports about gut microbiome of psoriasis patients, and whether the diversity of bacteria in genus level is decreased or not is still unclear. Moreover, it is not yet known if these gut bacteria would be the cause of the inflammation or the result of the inflammation. We analyzed the gut microbiome of the inflammatory skin model mouse (keratinocyte-specific caspase-1 transgenic (Kcasp1Tg) mouse), by analyzing the 16S rRNA gene. Staphylocuccus aureus and Streptococcus danieliae were abundant in Kcasp1Tg mouse fecal microbiome. These dominant bacteria as well as recessive control bacteria were orally administrated to antibiotic-treated wild type mice, and set up imiquimod-induced psoriasis-like skin inflammation model. The skin inflammation including ear thickness and histopathological findings was analyzed. The exacerbated skin lesions with the elevated levels of TNF-α, IL-17A, IL-17F, and IL-22 were observed in Staphylocuccus aureus and Streptococcus danieliae administrated groups. Our finding suggests that there is affinity between skin inflammation severity and certain gut bacteria leading to a vicious cycle: skin inflammation populates certain gut bacteria which itself worsens the skin inflammation. This is the first report on Staphylocuccus aureus and Streptococcuus danieliae effects in vivo. Not only treating the skin lesion but also treating the gut microbiome could be the future key treatment for inflammatory skin disease such as psoriasis.

## 1. Introduction

Next generation sequencing has paved the way for the analysis of the microbiome of mammals through the targeting of the 16S rRNA gene. Microbiota has been reported to influence the immune system of the host and cases of dysbiosis among these are known to be related with diseases with chronic inflammatory backgrounds [1]. Psoriasis is a common skin disease affecting a small percent of the world population [2]. Clinical features of psoriasis are known to include both cutaneous and extracutaneous manifestations due to the systemic inflammation associated with the disease. Accordingly, the interleukin 17 (IL-17) and tumor necrosis α (TNF-α) inflammatory cytokines produced in the skin lesion have been shown to be pathogenically important in the etiology of psoriasis [3].

In recent years, the skin and gut microbiome of patients with psoriasis have been investigated. The skin microbiome of psoriatic lesions has been reported to show an abundance of strains belonging to the *Streptococcus* genus and a meagerness of those of the *Propionibacterium* genus compared with those of patients with normal skin [4]. Atopic dermatitis, another chronic inflammatory skin disease is known to be colonized by members of the *Staphylococcus* genus, likely aggravating the disease by acting as superantigen [5]. On the other hand, the numbers of *Staphylococcus* genus strains in the skin of patients with psoriasis were demonstrated to be significantly lower in several reports [6,7], but higher in another case [8]. The gut-associated lymphoid tissue (GALT) is known to be the active center of systemic immune responses in the intestine. Respectively, stimulation through pattern-recognition receptors by the bacterial flora in the intestine has been suggested to be inevitable for the development of GALT [9]. In regard to cases of dysbiosis among members of the gut microbiome, research has shown that these might be responsible for inducing a number of diseases including inflammatory bowel disease, non-alcoholic steato-hepatitis, and Parkinson’s disease. However, only a few studies have been so far reported regarding the gut microbiome of patients with psoriasis. The gut microbiome of patients with psoriatic arthritis and psoriasis was demonstrated to exhibit less diversity compared with healthy controls [10]. More specifically, it was reported that the numbers of representatives in the *Actinobacteria* and *Bacteroidetes* phyla levels, as well as the numbers of *Parabacteroides, Coprobacillus,* and *Ruminococcus* genera, were decreased in the gut of patients with psoriasis [10]. On the contrary, Codoner et al. demonstrated that patients with psoriasis showed an increased diversity in their gut microbiome compared with the healthy population. An increase in the numbers of *Faecalibacterium, Akkermansia*, and *Ruminocuccus genera* and a decrease in the *Bacteroides* genus were also characteristic findings in the gut microbiome of patients with psoriasis [11]. Although these studies have produced conflicting results, the reported differences in the gut microbiome between healthy controls and patients with psoriasis remain significant, further suggesting the possible relation between gut dysbiosis and psoriasis. However, we could not specify from this analysis whether the dysbiosis is caused by the skin inflammation or if it is the cause of inflammation. Antibiotic treatment of gut of mice in an imiquimod-induced psoriasis mouse model was shown to lead to the reduced activation of T-helper 17 cells (Th17), resulting in milder inflammation of the skin [12]. The cause of the positive effect of the antibiotic-treated mouse toward the inflammation of the skin was not clear, but the involvement of the microbiome skin-gut axis was suggested.

Here we made the hypothesis that certain bacterial species present in the gut of an inflammatory skin disease mouse model were deeply associated with the manifested inflammation of the skin, and transplantation of these bacteria would worsen the symptoms in the skin of another inflammatory mouse model. Our research goal was to detect certain bacteria in the gut microbiome which might exacerbate the inflammation of the skin and thus contribute to the development of new treatment approaches targeting the skin-gut axis in patients with psoriasis.

## 2. Results

### 2.1. Abundance of Staphylococcus aureus and Streptococcus danieliae in the Gut Microbiome of keratinocyte-specific caspase-1 transgenic (Kcasp1Tg) Mice

We first examined the influence of the inflammation of the skin on the gut microbiome by analyzing the stools of Kcasp1Tg and wild type (WT) littermate mice. Analysis of the 16S rRNA gene using the Miseq system provided results down to the genus level of bacteria. The *Staphlococcus*, *Streptococcus*, *Prevotella*, and *Odoribacter* were the top four genera demonstrated to be significantly abundant in the intestine of Kcasp1Tg mice. We also used the Gridion X5 system to achieve analysis of the microbiome down to the species level. Regarding the *Prevotella* and *Odoribacter* genera, we did not detect any differences in the species level between the Kcasp1Tg and WT mice. In the case of the *Staphylococcus* genus, *Stapylococcus aureus* (SA) was shown to be the most abundant species in Kcasp1Tg mice, accounting for 93% of the genus. Likewise, regarding the *Streptococcus* genus, *Streptococcus danieliae* (SD), accounting for 92% of the representatives of the genus, was demonstrated to be the most ample species in Kcasp1Tg mice.

As a result, SA and SD were the two bacterial species exhibited to inhabit the intestine of Kcasp1Tg mice significantly more than that of WT littermate mice (Figure 1).

### 2.2. Oral Administration of SA and SD Worsened Skin Inflammation

To investigate the influence of these two bacterial species (SA and SD) in the inflammation of the skin, we tested the effect of orally administrating them to healthy C57BL/6 mice. To this end, we initially set to reduce the bacteria in the gut of healthy C57BL/6 mice by adding an antibiotic cocktail into their drinking water, as previously described [13,14,15]. After 10 days of administering antibiotic cocktail-containing water, each of the two bacterial species was orally gavaged for a period of 5 days, followed by application of imiquimod onto the shaved back and right ear of mice for 5 days to induce psoriasis-like inflammation of the skin. The *Bacteroides uniformis* (BU) and *Clostridum scindens* (CS) species were used as controls. These two bacterial strains were selected because they were detected in the gut microbiome of both Kcasp1Tg and WT mice, and constituted less than 0.01% of all bacteria, not showing any differences in abundance. A diagram of the study design is shown in Figure 2.

We measured the thickness of the right ear of mice during the 5-d period of application of imiquimod. Skin samples were collected under anesthesia at 72 and 120 h after the first application. Both SA- and SD-fed mice were shown to exhibit a significantly increased thickness on their ears compared to BU- and CS-fed mice at both 72 and 120 h (Figure 3a). At 120 h SA-fed mice were reported to exhibit the thickest ear among all groups. Phenotypically, SA- and SD-fed mice demonstrated the more severe lichenification, erythema, scale, and thickening of the skin (Figure 3b). Histopathological analysis of the back skin revealed the thickening of epidermis with hyperkeratosis, spongiosis, and micro abscess. The increased infiltration of mixed inflammatory cells into the papillary dermis was observed in both the SA and SD groups. At 72 h, SA- and SD-administrated mice were observed to exhibit micro abscess, mainly consisting of neutrophils (Figure 3c).

### 2.3. The Expression of TNF-α, IL-17A, IL-17F, and IL-22 Was Elevated in the Skin of SA- and SD-Administered Mice

We examined the mRNA expression levels of inflammatory cytokines related to psoriasis. At 72 h, the expression of TNF-α, IL-17A, IL-17F, and IL-22 showed significantly higher levels in SA- and SD-administered mice compared with those in BU- and CS-administered mice. Among the SA and SD groups, only the level of TNF-α was observed to be significantly higher in the SA relative to that of the SD group. At 120 h, the BU group was reported to exhibit the lower levels of TNF-α compared with the SA and SD groups. On the other hand, the CS group was demonstrated to exhibit higher levels of TNF-α compared with the SA and SD groups. As for IL-17A and IL-17F, both the SA and SD groups were observed to exhibit higher levels of the cytokines in comparison with the BU and CS groups, similar to the results obtained at 72 h. The relative mRNA level of IL-22 was shown to be higher in the SA and SD compared with that in the BU and CS groups (Figure 4).

## 3. Discussion

In patients with chronic inflammatory skin disorders, dysbiosis of both the skin and gut microbiome is known to be associated with the pathogenesis of the disease [7,16]. A previous report for patients with psoriasis showed a clear dysbiosis of their microbiome, although there was no particular bacteria detected from the phylum to genus level [17,18,19]. Therefore, there has been no report suggesting whether a particular bacteria habiting in the gut of patients with inflammatory skin disease might be the result of inflammation or could actually be the etiological factor for the worsening of the inflammation of the skin.

To investigate the influence of the gut microbiome in the inflammation of the skin, we first analyzed the gut microbiome of the Kcasp1Tg inflammatory skin disease mouse model. There was no significant perturbation in the phylum level, such as a difference in the *Firmicutes/Bacteroides* ratio reported in other skin inflammatory diseases [20,21,22]. Likewise, no differences were detected in the microbiome between Kcasp1Tg and WT mice neither in class, order, nor family level. However, four strains of bacteria were observed to be significantly abundant in Kcasp1Tg mice in the genus level (Figure 1). Further analysis of the 16S rRNA gene enabled us to analyze the microbiome down to the species level and revealed a higher percentage of *Staphylococcus aureus* (SA) and *Streptococcus danieliae* (SD) in the gut microbiome of Kcasp1Tg mice. No bacterial species belonging to the *Odoribacter* and *Prevotella* genera were detected or observed to grow significantly abundant in the Kcasp1Tg mice. The reason for this controversial result could be that as the level of investigation becomes more detailed, 16S rRNA sequencing detecting the bacteria becomes marginal.

In the *Staphylococcus* and *Streptococcus* genera, the SA and SD strains were shown to be dominating over 90 % of the respective genus level in the intestine of Kcasp1Tg mice. Our hypothesis was that bacteria colonizing the gut of the inflammatory skin disease model might be the cause of the inflammation of the skin. To test this hypothesis, we cultured these two bacterial species (SA and SD) and administrated them into WT antibiotic-treated mice with imiquimod-induced experimental psoriasis. As a result, both SA and SD were demonstrated to worsen the inflammation of the skin. In addition, the supplementation of SA and SD in the gut of WT mice without giving prior antibiotic treatment showed no significant difference in the ear thickness (Appendix A) nor cytokine levels compared to BU and CS mice at 72 h and 120 h (Appendix A).

Notably, SA is a known bacterial strain colonizing the gut of patients with atopic dermatitis [21,23]. Accordingly, SA has been reported to activate human mucosal-associated invariant T-cells, γδ T-cells, NK cells, as well as CD4(+) T–cells and CD8(+) T-cells in vitro, therefore explaining the flaring of the inflammation of the skin [24]. On the other hand, such observations have not been made yet in the case of SD. In particular, SD was first isolated from the cecum of TNF (deltaARE) Crohn’s disease mice [25]; however there have been no reports regarding the pathogenicity of this bacterial strain. We have since found that the inflammation of the skin might promote the growth of SD in the intestine, and colonization of the gut by SD itself has been shown to constitute an aggravation factor for inflammation of the skin.

Compared with BU mice, the relative mRNA levels of the TNF-α, IL-17A, IL-17F, and IL-22 cytokines were shown to be significantly increased in the skin lesions in SA and SD mice. With the exception of IL-22 at 72 h and TNF-α at 120 h, overall CS mice were reported to exhibit similar tendencies as those observed for BU mice. Macroscopic and histopathological findings revealed severe inflammation of the skin on SA and SD compared with BU and CS mice. The maximum level of cytokines is known to be expressed around 72 h and the phenotypic peek has been shown to occur around 120 h in imiquimod-induced dermatitis [26], consistent with our results. Of note, CS mice were observed to exhibit higher mRNA levels of TNF-α at 120 h. We assumed that the cytokine levels in SA and SD mice had already peaked at 120 h. The total expression level of cytokines during the treatment should have been higher in the SA and SD mice, therefore exhibiting a more severe phenotypic presentation.

The IL-17 and IL-22 cytokines have been considered as hallmarks of the inflammatory phenotype in patients with psoriasis. Compared with healthy controls, the skin of patients with psoriasis has been shown to contain increased levels of Th17 cells with IL-17A, IL-17F, and IL-22 proteins being abundantly produced [27,28]. The IL-17 cytokine is known to induce the expression of proinflammatory cytokines and activate neutrophils [29]. Both SA and SD mice were noted to exhibit increased infiltration and migration of neutrophils into the epidermis presenting a Munro’s micro-abscess-like phenotype. IL-22 has also been reported as an important cytokine of the IL-17/IL-23 axis of psoriasis, and elevated levels of IL-22 have been shown in the skin and serum of patients [30,31], produced by Th22, γδ T-cells, NK cells, and mast cells [3]. Accordingly, it was shown that if IL-22 was neutralized by antibodies, the development of psoriasis was prevented in the mouse model [32]. Multiple innate and adaptive lymphocytes are the center of the immune system in intestinal mucosa. Respectively, γδ T-cells and Th17 cells are known to be contributing to the host defense in the intestine, both being sources of IL-17 and IL-22 [1].

Antibiotic treatment in an imiquimod-induced model has been shown to result in milder dermatitis, due to a background of reduced IL-22+γδ T-cells in the skin as well as reduced Th17 cells in the gut [33]. In addition, those mice showed a systemic reduction of γδ T-cells and Th17 cells in the lymph nodes and spleen [12]. Several studies have demonstrated the existence of an interactive axis between the gut and skin [16,21,34]. We think that SA- and SD-administrated mice exhibited a positive reaction in the gut by increasing the activation of immune cells, such as Th17 cells, with the increased production of IL-17 and IL-22 aggravating as well the phenotype of psoriasis-like dermatitis. Our findings suggested an association between the gut microbiome and the inflammation of the skin; bacteria likely to propagate in the gut of patients with inflammatory skin disease would be the cause of inflammation of the skin. There have been some reports suggesting the use of probiotics for the treatment of psoriasis [17,35,36]. Probiotic treatments could be shown to be more effective if combined with antibacterial treatment, as antibiotics have been demonstrated to reduce the number of lymphocytes producing inflammatory cytokines in murine models. We identified two bacterial strains purified from the gut of Kcasp1Tg mice that were shown to result in the worsening of the inflammation of the skin. However, the efficacy by which these bacteria might lead to the worsening of a skin disease in actual human patients could vary. Hence, changing the whole microbiome through fecal transplantation could as well be a future option in the management of skin diseases. For instance, such approaches have been shown to be efficacious in the treatment of inflammatory bowel disease [37,38] and infections by *Clostridium difficile* [39].

The intriguing fact that certain bacteria in the gut have been shown to be affecting the condition of the skin indicates the importance of further research regarding the interplay between the gut microbiome and inflammatory skin diseases, and emphasizes the value of treating both the skin and the gut microbiome for the management of inflammatory skin diseases.

## 4. Materials and Methods

### 4.1. Animals

The keratin-14 driven caspase-1 transgenic (Kcasp1Tg) mice were used in this study as a spontaneous inflammatory dermatitis model [40]. The littermate mice were used as controls. Seven-week-old C57BL/6N wild type female mice were purchased from Japan SLC Inc. (Shizuoka, Japan) and housed in a pathogen-free environment at the experimental animal center of Mie University. Mice were kept in an air-conditioned room at 22 ± 2 °C and 50 ± 10% humidity with a 12 h light-dark cycle and had free access to standard laboratory food (DC-8, CLEA Japan Inc., Tokyo, Japan) and distilled water until the age of 8 wk. Animals were cared for according to the ethical guidelines, and all experimental protocols were approved by the Mie University Board Committee for Animal Care and Use (#22-39-4, approved on 03 Dec 2018).

### 4.2. 16S rRNA Sequencing and Analysis of Gut Microbiome

We collected stool samples from 16-wk-old female Kcasp1Tg mice and wild type (WT) littermate mice. DNA was extracted using the MPure Bacterial DNA extraction kit (MP Biomedicals, Irvine, CA, USA). Sequencing was performed on the Illumina’s MiSeq platform (Illumina, San Diego, CA, USA) using a 2 × 300 bp run. Sequencing data were processed using the QIIME (Quantitative Insights Into Microbial Ecology) bioinformatics pipeline. Analysis of our next generation sequencing data revealed the microbiome to the genera level. Stool samples of Kcap1Tg mice were also analyzed using the GRIDion X5 system by Oxford Nanopore Technologies (Oxford Science Park, Oxford, UK), which enabled us to analyze our data to the species level. Data were analyzed using the USEARCH software and the RDP_16S_V16_sp database.

### 4.3. Bacterial Samples

First the skin of Kasp1Tg mice was sanitized by spraying and wiping with ethanol (Nacalai tesque, Kyoto, Japan). Then, freshly excreted stool samples were transferred into 1.5 mL tubes. Each stool sample was suspended in 1 mL distilled water, while 10 μL was spread on Mannitol salt agar containing egg yolk (Kyokuto, Tokyo, Japan). *Staphylococcus aureus* (SA) was identified according to its morphological growth on selective medium, Gram staining (Muto pure chemicals, Tokyo, Japan), as well as by performing catalase and oxidase tests. SA was cultivated on Trypticase soy agar (BD, Franklin Lakes, NJ, USA) in aerobiotic conditions for 48 h at 37 °C. *Streptococcus danieliae* (SD) was purchased from DSMZ (Leipzig, Germany), and cultured in Colombia sheep blood agar (BD) under 5% CO_2_ aerobiotic conditions for 48 h at 37 °C. As negative controls, we selected 2 bacterial species isolated in similar percentages in the microbiome of both wild type and Kcasp1Tg mice. These 2 bacteria were *Clostridium scindens* (CS) and *Bacteroides uniformis* (BU). Both, CS and BU (ATCC, Manassas, VA, USA) were cultured in TSA 5% sheep blood agar (BD) under 5% CO_2_ anaerobic conditions for 48 to 72 h at 37 °C.

### 4.4. Experimentally Induced Psoriasis

C57BL/6N littermate mice were treated with antibiotics dissolved in their drinking water ad libitum for 10 d. A combination of 1 g/L ampicillin (Sigma Aldrich, St. Louis MO, USA), 1 g/L neomycin sulfate, 1 g/L metronidazole, and 500 mg/L vancomycin (Funakoshi, Tokyo, Japan) was administered as previously described [13,41]. Respectively, 5 × 10^5^/CFU bacteria were suspended in 200 μL phosphate buffered saline and orally gavaged with a 3.5 cm length feeding tube (AS ONE co., Osaka, Japan) into the stomach of mice daily for a total of 5 days. On the 3rd day of gavage, the back skins were shaved with electric razor (Natsume Seisakusho, Tokyo, Japan). From the 5th day onwards, 60 mg of imiquimod cream (Mochida Pharmaceutical, Tokyo, Japan) was applied on the back and ear of WT mice for 5 consecutive days. Every day the thickness of the ear was measured by a micrometer (Mitsutoyo, Kanagawa, Japan). Subsequently, 5 mm diameter skin samples were collected for the purpose of histology and gene analysis at 72 and 120 h since the first application of imiquimod.

### 4.5. Quantitative Real-Time PCR

RNA was extracted from skin tissue samples using the TRI reagent (Molecular research center, Cincinnati, OH, USA). Then, cDNA was prepared using the High-Capacity RNA-to-cDNA™ Kit (Thermo Fisher Scientific, Waltham, MA, USA). The mRNA levels of TNF-α, IL-17A, IL-17F, IL-22, and GAPDH were measured by real-time quantitative PCR analysis using a LightCycler 96 System (Roche, Basel, Switzerland). The thermal cycle conditions were pre-incubation for 2 min at 50 °C followed by 10 min at 95 °C for the initial activation. Amplification was 40 cycles of denaturation for 15 s at 95 °C, and primer annealing for 60 s at 60 °C. The following primers obtained from TaqMan (Thermo Fisher Scientific) were used for PCR reactions: TNF-α, Mm00443258_m1; IL-17A, Mm00439618_m1; IL-17F, Mm00521423_m1; IL-22, Mm01226722_g1; GAPDH, Mm99999915_g1.

### 4.6. Statistical Analysis

Statistical analyses were performed using the GraphPad Prism 6 (GraphPad Software, La Jolla, CA, USA). *p* values < 0.05 were considered to be statistically significant in a two-tailed *t*-test. The Mann–Whitney test was used when comparing 2 groups, and the Kruskal–Wallis test when undergoing multiple comparisons.

### 4.7. Histological Analysis

Skin specimens were obtained 72 and 120 h after the first application of imiquimod. Specimens were fixed in 10% buffered neutral formaldehyde (Nacalai tesque, Kyoto, Japan) embedded in paraffin, and sections cut into 4 μm were stained with hematoxylin and eosin (Genostaff, Tokyo, Japan). Sections were examined by light microscopy (OLYMPUS, Tokyo, Japan).

## Figures and Tables

**Figure 1 ijms-21-03303-f001:**
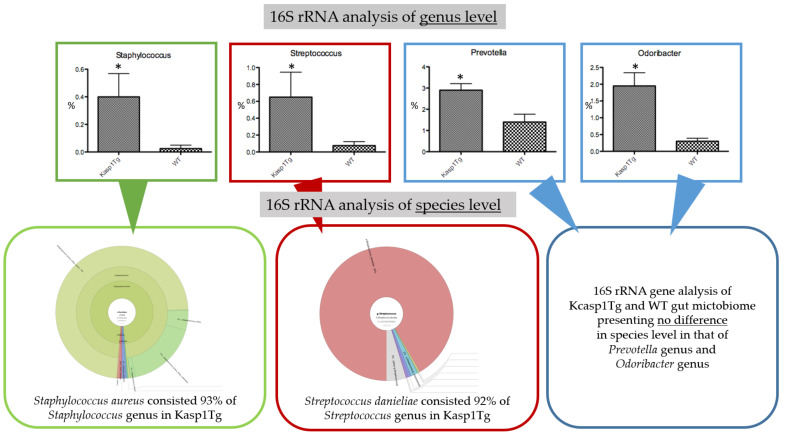
Analysis of 16S rRNA of the gut microbiome reveals a significant increase of the *Staphylococcus*, *Streptococcus*, *Prevotella*, and *Odoribacter* genera. **p* < 0.05. All values were analyzed using the Mann–Whitney test. Further analysis down to the species level revealed that *Staphylococcus aureus* and *Streptococcus danieliae* were the most abundant strains in the *Staphylococcus* and *Streptococcus* genera, respectively.

**Figure 2 ijms-21-03303-f002:**
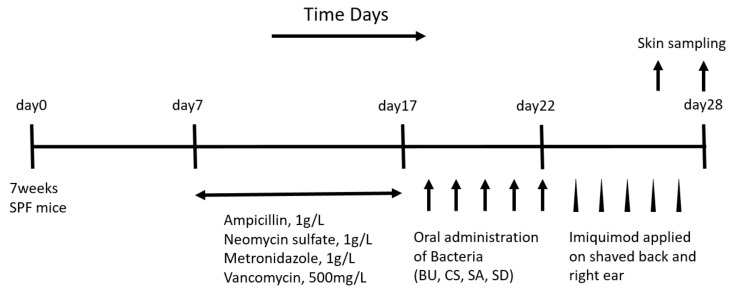
Experimental design.

**Figure 3 ijms-21-03303-f003:**
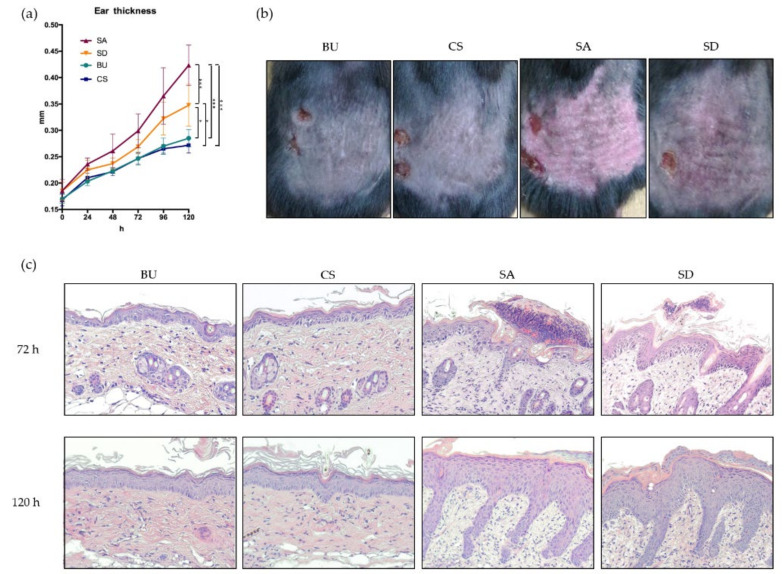
*Stapylococcus aureus* (SA)- and *Streptococcus danieliae* (SD)-administrated mice exhibit aggravated inflammation of the skin in an imiquimod-induced dermatitis model. (**a**) Phenotypically, SA- and SD-fed mice have significantly thicker ears compared with *Bacteroides uniformis* (BU)- and *Clostridum scindens* (CS)-fed mice over time. **p* < 0.05, ****p* < 0.001. (**b**) SA- and SD-fed mice exhibit stronger erythema, scales, and hyperplasia compared with BU- and CS-fed mice. (**c**) Hematoxylin and eosin staining of the back skin shows thickened epidermis, hyperkeratosis, spongiosis, micro abscess, and increased infiltration of mixed inflammatory cells in SA- and SD-fed mice compared with BU- and CS-fed mice. Original magnification ×200.

**Figure 4 ijms-21-03303-f004:**
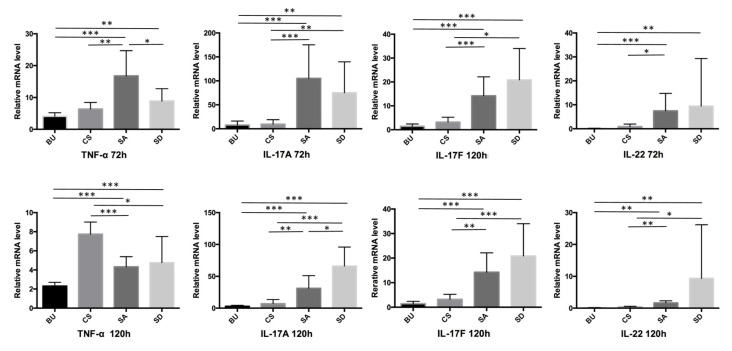
The expression of TNF-α, IL-17A, IL-17F, and IL-22 genes is elevated in SA- and SD-fed mice compared with controls. **p* < 0.05, ***p* < 0.01, ****p* < 0.001.

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
