# Peer review of "The Role of Gut Microbiome in Psoriasis: Oral Administration of Staphylococcus aureus and Streptococcus danieliae Exacerbates Skin Inflammation of Imiquimod-Induced Psoriasis-Like Dermatitis"

_ijms, 2020, doi:10.3390/ijms21093303_

Round 1
Reviewer 1 Report
Fascinating research highlighting the skin gut axis. The outcomes are very promising and raise the need for extending the research on humans, not only mouse models. The authors reported that treating gut microbiome could be a new therapeutic target in the future in order to improve the psoriatic lesions. Great job.
Author Response
Responses to the comments of Reviewer #1
Comments to the Author: Fascinating research highlighting the skin gut axis. The outcomes are very promising and raise the need for extending the research on humans, not only mouse models. The authors reported that treating gut microbiome could be a new therapeutic target in the future in order to improve the psoriatic lesions. Great job.
Response: We appreciate your review and are thankful for your assessment of our work. We believe that our research will aid future psoriasis treatments.
Reviewer 2 Report
The authors elucidate the role staphylococcus, when delivered orally, play a role in chemical-induced psoriasis in mice. The conclusions are that SA and SD exacerbate psoriasis using transgenic and chemical-induced psoriasis models. Specific mechanisms remain elusive and the paper is largely descriptive but important observations were seen. Some aspects of the paper that need to be clarified include:
1. It is unclear what controls were used in the transgenic mouse studies with 16S - did the authors use littermate or parental non-transgenic strains for 16S sequencing? And at what ages or age range were they assessed?
As a note, 16S sequencing is very marginal at determining species-level information - as such metagenomics is better and this should at least be commented on.
2. Mechanistically, the cytokine assessments are very basic - there is the elevation of TNF and IL17 but also IL22 - here IL22 at least in the intestine acts top fortify the barrier and acts in opposition to IL17 - so a bit deeper exploration as to what immune pathways are being triggered for a distal response in the skin needs to explored. Some attempt to determine which immune subsets are responsible for the skin effects after dosing with SA etc.. would be an important advance.
3. Experiments for repletion with SA and SD were performed - inverse experiments to determine cause and effect might also be important such as antibiotic depletion in transgenic mice followed by mono inoculation studies with SA and SD? Also, what happens to non-antibiotic depleted mice when dosed with SA and SD - while they may not colonize do they increase chemical-induced psoriasis?
4. In the experiments proposed in normal C57B/L6 mice, was 16S seq used to monitor each stage of the experimental setup--like pre-and post-antibiotic - if so this data should be shown at least in supplementary.
5. Method details are important--for example, the histology section is vague --what microsections were used etc... details throughout are important.
Author Response
Responses to the comments of Reviewer #2
Comments to the Author: The authors elucidate the role staphylococcus, when delivered orally, play a role in chemical-induced psoriasis in mice. The conclusions are that SA and SD exacerbate psoriasis using transgenic and chemical-induced psoriasis models. Specific mechanisms remain elusive and the paper is largely descriptive but important observations were seen. Some aspects of the paper that need to be clarified include:
- It is unclear what controls were used in the transgenic mouse studies with 16S - did the authors use littermate or parental non-transgenic strains for 16S sequencing? And at what ages or agerange were they assessed? As a note, 16S sequencing is very marginal at determining species-level information - as such metagenomics is better and this should at least be commented on.
Response: Thank you for your detection. Yes, the controls used for 16S were 16-weeks-old littermate mice housing together with transgenic mouse. We have employed state-of-the-art system to analyse 16S sequence, but we have added a comment that determining species-level 16S sequencing is marginal.
- Mechanistically, the cytokine assessments are very basic there is the elevation of TNF and IL17 but also IL22 - here IL22 at least in the intestine acts top fortify the barrier and acts in opposition to IL17 - so a bit deeper exploration as to what immune pathways are being triggered for a distal response in the skin needs to explored. Some attempt to determine which immune subsets are responsible for the skin effects after dosing with SA etc.. would be an important advance.
Response: Thank you for the profound advice about the cytokine pathways and roles. It is true that, while IL-17 mostly reacts as pro-inflammatory cytokine, IL-22 is more known to be homeostatic cytokine but occasionally works as pro-inflammatory cytokine. As you pointed out, IL-22 shows protective role in the intestine, such as the idiopathic inflammatory bowel diseases (IBDs). In IBDs both IL-17 and IL-22 are increased in the mucosa of the intestine. Blocking the IL-17 didn’t made the inflammation better (Gut. 2012 Dec;61(12):1693-700.), but treating with IL-22 improved the bowel inflammation and IL-22-/- mice showed exacerbation of the inflammation (Immunity. 2008 Dec 19;29(6):947-57.). On the other hand, IL-22 producing transgenic mice show psoriasis-like skin inflammation (J Mol Med (Berl). 2009 May;87(5):523-36.). We therefore assumed that the increased production of IL-22, while protecting the intestinal inflammation, would not contradict the fact that it worsens the psoriasis-like skin inflammation. We appreciated your great comments.
- Experiments for repletion with SA and SD were performed inverse experiments to determine cause and effect might also be important such as antibiotic depletion in transgenic mice followed by mono inoculation studies with SA and SD? Also, what happens to non-antibiotic depleted mice when dosed with SA and SD - while they may not colonize do they increase chemical inducedpsoriasis?
Response: Thank you for the further suggestion for our experiments. For the first question, the inflammatory model mouse Kcasp1Tg showed strong skin inflammation which masked the effect of experimental psoriasis. Therefore we could not obtain an adequate result. Kcasp1Tg is also fragile against stress compared to wild type mice, and Kcasp1Tg could not survive the experiment with antibiotic administration for 10 days.
For the second question, we performed an additional experiment. We administrated SA and SD to 8 weeks old wild type mice (n=6) without the antibiotic treatment and obtained the data. Underneath are the figures for the ear thickness and RT-PCR results. SA’ and SD’ are meaning the newly supplemented data for non-antibiotic treated mice. Statistical difference between BU and CS (control group) vs SA’ and SD’ are shown on the graph (* p < 0.05, ** p < 0.01). At 72hrs, SA’ and SD’ presented significantly lower TNF, IL-17A, IL-17F than BU and CS. This data indicates that SA and SD did not colonize in the gut as well as antibiotic treated mice. At 120hrs, TNF- α was higher for SA’ than BU but SD’ was lower than CS. We assume that the cytokine levels already peaked out at this time period resulting into these data.
Ear thickness were not different between controls and SA’ and SD’ at 72hrs and 120hrs. Although the ear thickness of SA’ and SD’ at 0hrs were significantly thicker than the controls (p < 0.01), we think this was due to the dehydration of antibiotic treated mice. When mice were drinking antibiotic water, they drank less than half of the usual drinking water.
As a conclusion, we imagine that SA and SD colonize better when pre-antibiotic treatment were given and exacerbates skin inflammation. These data has been added in the supplementary figure. We appreciated your comments.
- In the experiments proposed in normal C57B/L6 mice, was 16S seq used to monitor each stage of the experimental setup—like pre-and post-antibiotic - if so this data should be shown at least insupplementary.
Response: Unfortunately, we just used the 16S rRNA gene sequencing to figure out the difference of gut microbiomes for controls and inflammatory model mice and did not monitor the microbiomes pre-and post-antibiotic treatment. We understand the importance to compare the microbiome at each stage and in future studies we would like to perform it.
- Method details are important--for example, the histology section is vague --what microsections were used etc... details throughout are important.
Response: Thank you for your detection. We revised the addressed sections and added further details.
Round 2
Reviewer 2 Report
acceptable